# Identification of the 14-3-3 Gene Family in Bamboo and Characterization of *Pe14-3-3b* Reveals Its Potential Role in Promoting Growth

**DOI:** 10.3390/ijms231911221

**Published:** 2022-09-23

**Authors:** Dong Guo, Chenglei Zhu, Kebin Yang, Yan Liu, Xiaoyan Xiao, Ziyang Li, Zhimin Gao

**Affiliations:** Key Laboratory of National Forestry and Grassland Administration/Beijing for Bamboo and Rattan Science and Technology, Institute of Gene Science and Industrialization for Bamboo and Rattan Resources, International Center for Bamboo and Rattan, Beijing 100102, China

**Keywords:** bamboo, *14-3-3* gene, phylogenetic analysis, co-expression, promoting growth

## Abstract

The 14-3-3 protein family plays an important role in regulating plant growth and development. The genes of the 14-3-3 family have been reported in multiple species. However, little is known about the 14-3-3 gene family in bamboo. In this study, a total of 58 genes belonging to the 14-3-3 family were identified in three representative bamboo species, i.e., *Olyra latifolia*, *Phyllostachys edulis*, and *Bonia amplexicaulis*, whose encoding proteins were grouped into ε and non-ε groups by phylogeny analysis with 14-3-3 proteins from *Arabidopsis thaliana* and *Oryza sativa*. The 14-3-3s had diverse gene structures and motif characteristics among the three bamboo species. Collinearity analysis suggested that the genes of the 14-3-3 family in bamboo had undergone a strong purification selection during evolution. Tissue-specific expression analysis showed the expression of *Pe14-3-3*s varied in different tissues of *P*. *edulis*, suggesting that they had functional diversity during growth and development. Co-expression analysis showed that four *Pe14-3-3*s co-expressed positively with eight ribosomal genes. Yeast two-hybrid (Y2H) assays showed that Pe14-3-3b/d could interact with Pe_ribosome-1/5/6, and qPCR results demonstrated that *Pe14-3-3b*/*d* and *Pe_ribosome-1*/*5*/*6* had similar expression trends with the increase in shoot height, which further confirmed that they would work together to participate in the shoot growth and development of bamboo. Additionally, the transgenic *Arabidopsis* plants overexpressing *Pe14-3-3b* had longer roots, a larger stem diameter, an earlier bolting time and a faster growth rate than wild-type *Arabidopsis*, indicating that *Pe14-3-3b* acted as a growth promoter. Our results provide comprehensive information on 14-3-3 genes in bamboo and highlight *Pe14-3-3b* as a potential target for bamboo improvement.

## 1. Introduction

The 14-3-3 protein family was originally discovered and isolated from bovine brain [1]. In plants, the first 14-3-3 protein was isolated from *Arabidopsis thaliana* and named GF14 [2]. Subsequently, 14-3-3 proteins were detected in many plant species and named GF or GRF [2,3]. The latest research clearly confirmed that 14-3-3s were widely involved in regulating plant growth and development, such as cell elongation and cell division, seed germination and seed dormancy [4]. For example, in cotton (*Gossypium hirsutum*), 14-3-3 proteins were involved in cell elongation, which were highly induced during the rapid cell-elongation phase in fiber development [5]. In rice (*Oryza sativa*), the overexpression of *Os**GF14c* delayed flowering, while its knockout mutants displayed early flowering [6]. In soybean (*Glycine soja*), *GsGF14o* overexpression caused deficits in root-hair formation and development [7]. In *A. thaliana*, the *14-3-3μ* T-DNA insertion mutant, *14-3-3μ-1*, had shorter roots than the wild type [8], and the overexpression of *AtGRF9* favored the allocation of shoot carbon to roots and enhanced proton extrusion in the root-elongation zone and the root-hair zone [9]. In cassava (*Manihot esculenta*), 14-3-3 proteins participated in carbohydrate metabolism and starch accumulation during cassava root tuberization, and an overexpression of a cassava 14-3-3 gene increased the sugar and starch content of older leaves in *A*. *thaliana* [10]. However, the overexpression of *Ta14-3-3* in *A*. *thaliana* inhibited the growth of primary roots [11]. Additionally, a prominent role of 14-3-3 proteins their involvement in regulating plant immunity against pathogens to influence plant growth and development. For example, a *G*. *hirsutum* 14-3-3 was rapidly expressed in response to *Verticillium dahliae* in a cultivar with enhanced wilt resistance [12], and in *O*. *sativa*, OsGF14c positively regulated immunity by modulating the protein homoeostasis of the GRAS protein OsSCL7 [13]. These results indicated the diverse function of 14-3-3 genes.

Furthermore, the interaction between 14-3-3 proteins and other proteins during plant growth and development has been reported in recent years. For instance, the GhGRF3/6/9/15 in cotton interacted with GhFT and GhFD to inhibit flowering, while GhGRF14 interacted with GhFT and GhFD to promote flowering [14]. In apples (*Malus domestica*), 14-3-3 proteins (MdGF14a, MdGF14d, MdGF14i, and MdGF14j) participated in the development of flowering through interaction with MdTFL1 and MdFT [15]. In rice roots, 14-3-3 proteins (GF14b, GF14c, GF14f and GF14e) interacted with root hair development-related proteins involved in regulating root growth [5]. In *Arabidopsis*, 14-3-3ε isoforms interacted with four seed storage proteins (At4g27170.1, At4g28520.1, At4g28520.2 and At1g07750) in the process of seed development [16]. A large number of 14-3-3 proteins and ribosomal proteins were found during the development of bud differentiation in *Pinus pinaster*, and the numbers increased with bud maturation [17]. It was found that 14-3-3 proteins could interact with p90 ribosomal S6 kinase (RSK), negatively regulating its activity to affect growth [18]. In *Arabidopsis*, most proteins identified in protein synthesis during seed development were ribosomal-related and could interact with the 14-3-3 isoforms χ and ε [16]. Additionally, studies have shown that 14-3-3 proteins could bind ADF4 (one protein of the actin-depolymerizing factor family) in regulating hypocotyl growth [19]. These results indicated that 14-3-3 proteins functioned together with other proteins during plants growth and development.

Moso bamboo (*Phyllostachys edulis*) is the most widely planted bamboo species in China with a large yield [20]. Moso bamboo is characterized by fast growth, with the ability to grow more than 1.0 m per day during the rapid-growth period and reaching its maximum height within one and a half months [21]. Before bamboo shoots were excavated, the number of nodes of bamboo culm were determined, and the growth of bamboo was mainly demonstrated by the growth of internodes after the excavation [22]. Cell division and cell elongation played an important role in the rapid growth of bamboo shoots [23], which were regulated by a series of factors, including hormones [18], proteins [24], transcriptional factors [25] and so on. Although 14-3-3 proteins play an important role in cell division and cell elongation during plant growth and development as an important regulatory protein [5,26], the role of the 14-3-3 family in bamboo is still unclear. In this study, members of the 14-3-3 gene family were identified in the genomes of *Olyra latifolia*, *P*. *edulis* and *Bonia amplexicaulis*, and further comprehensive analyses of molecular characteristics and the evolutionary relationship of 14-3-3 genes were conducted. Moreover, the function of *Pe14-3-3b* was validated by ectopic expression in *A. thaliana*, and a yeast two-hybrid (Y2H) assay. The results provided a comprehensive understanding of the 14-3-3 gene family in bamboo and revealed that *Pe14-3-3b* is a candidate gene for the improvement of bamboo breeding.

## 2. Results

### 2.1. Identification, Molecular Characteristics and Phylogenetic Analysis of 14-3-3 Family Members in Bamboos

A total of 58 members of the 14-3-3 gene family were identified (Appendix A) in *O*. *latifolia*, *P*. *edulis* and *B*. *amplexicaulis*. Among the 14-3-3 genes of *P*. *edulis*, 15 encoding protein members with a complete 14-3-3 domain were named *Pe14-3-3a* to *Pe14-3-3o*; the remaining 21 members with an incomplete 14-3-3 domain were named as *Pe14-3-3like1* to *Pe14-3-3like21* (Appendix A). Similarly, those eight genes identified in *O*. *latifolia* and 14 genes identified in *B*. *amplexicaulis* were named *Ol14-3-3a* to *Ol14-3-3h* and *Ba14-3-3a* to *Ba14-3-3n*, respectively (Appendix A). The prediction of proteins encoded by *Pe14-3-3*s showed that they had different physical and chemical properties, including different protein lengths, ranging from 106 aa (Pe14-3-3like8 and Pe14-3-3like15) to 287 aa (Ba14-3-3l), and different molecular weights (MW), ranging from 11.77 kDa (Pe14-3-3like15) to 45.68 kDa (Pe14-3-3like10). Additionally, the theoretical isoelectric points (pI), subcellular localization and aliphatic index were also predicted to be different (Appendix A). To understand the evolutionary relationships among the 14-3-3 proteins, an unrooted phylogenetic tree was constructed using Pe14-3-3a~Pe14-3-3o and those of other four plant species (*A*. *thaliana*, *B*. *amplexicaulis*, *O*. *latifolia*, and *O*. *sativa*) (Appendix A). The results (Figure 1) showed that all the 14-3-3 proteins were obviously divided into ε and non-ε groups, including 12 and 46 members, respectively.

Furthermore, the gene structure analysis (Appendix A) showed that *Pe14-3-3a~Pe14-3-3o* had 4~7 exons with differences in distribution and length, while *Pe14-3-3like1~Pe14-3-3like21* had 3~12 exons. None of them had untranslated regions (UTR) structures except *Pe14-3-3ike7*, *Pe14-3-3like8*, and *Pe14-3-3like12*. Motif analysis (Appendix A) demonstrated that the number of motifs in Pe14-3-3a~Pe14-3-3o ranged from 9 to 11, and their positions were roughly the same: all the members shared motif 1 to motif 6. While the motifs in Pe14-3-3like1~Pe14-3-3like21 ranged from 1 to 9, the location of each motif also varied greatly: Pe14-3-3like1 contained nine motifs, while Pe14-3-3like18 only contained motif 6 (Appendix A). Differentially, the 14-3-3 members in *O*. *latifolia* and *B*. *amplexicaulis* showed different gene structures and motif characteristics with those members of *P*. *edulis*, whose members had 4~7 exons and 7 to 11 motifs (Appendix A).

### 2.2. Scaffold Localization and Collinearity Analysis of 14-3-3 Genes

The genome location analysis revealed that 14-3-3 genes were unevenly distributed on the scaffolds. The identified 14-3-3 genes were located in 8, 20 and 13 scaffolds of *O*. *latifolia*, *P*. *edulis*, and *B*. *amplexicaulis*, respectively. To fully elucidate the genetic evolution in bamboo, three typical circular maps were constructed by comparing 14-3-3 genes of *P*. *edulis* and other three plants (*O*. *sativa*, *O*. *latifolia* and *B*. *amplexicaulis*). The intraspecific collinearity results showed that 15 *14-3-3* orthologous gene pairs were identified in *P*. *edulis*, and interspecific collinearity results revealed that 21, 27, and 40 *14-3-3* orthologous gene pairs between *P*. *edulis* and *O*. *sativa*, *O*. *latifolia* and *B*. *amplexicaulis* (Figure 2a–c, Appendix A) were identified, respectively, suggesting that *Pe14-3-3*s may have existed before the differentiation of the other three species. To explore the selection pressure of the 14-3-3 genes in these four plants, the *Ka* (*non-synonymous substitution*)/*Ks* (*synonymous substitution*) ratios of duplicated gene pairs were calculated. All *Ka*/*Ks* ratios of *14-3-3* orthologous gene pairs within *P*. *edulis*, and between *P*. *edulis* and the three other species, were less than 1 (Figure 2d, Appendix A), indicating that these 14-3-3 genes had undergone a purifying selection during evolution [27].

### 2.3. Tissue Expression Patterns of Pe14-3-3s in Moso Bamboo

To reveal the role of *Pe14-3-3*s, we investigated their spatiotemporal expression patterns based on the published transcriptome data [28], and an expression heatmap was drawn (Figure 3). The results showed that *Pe14-3-3*s exhibited a tissue-specific expression and were further divided into three groups. In group I, 16 *Pe14-3-3*s were nearly highly expressed in all the samples of moso bamboo and showed different expression patterns. For example, some members presented a constitutive expression pattern, such as *Pe14-3-3g* and *Pe14-3-3i*, suggesting that they played an important role in different tissues during growth and development. Some members, such as *Pel4-4-3b*/*c*/*a*/*d*/*f*/*like1*, had similar patterns both in roots and shoots at different growth stages, and were highly expressed in the top part of the roots, shoots and buds, suggesting that these genes played an important role in different growth stages, especially involved in the development of actively differentiated tissues. Meanwhile, *Pe14-3-3k* and *Pe14-3-3l* were highly expressed in specific tissues, including rhizome roots with a 0.5 cm length, the tip of bamboo shoots and the buds, suggesting that they might function in particular tissues. Of 36 genes, 5 *Pe14-3-3*s belonged to group II, which was only expressed in certain tissues and had a very low expression level, such as *Pe14-3-3like7*, which was expressed relatively higher in rhizome roots with a 10 cm length and leaf sheath. In group III, the remaining 14 *Pe14-3-3*s, except *Pe14-3-3n*, could hardly be detected in all tissues, suggesting that they may be functionally redundant genes.

### 2.4. Co-expression Analysis of Pe14-3-3s

To further investigate the co-expression genes of *Pe14-3-3*s and the metabolic pathways they might be involved in, the prediction of co-expression analysis was performed with 16 *Pe14-3-3*s (group I family members), which were nearly expressed in all detected tissues. A total of 271 genes were found to be co-expressed with these *Pe14-3-3*s, in which the genes that participated in the ribosome pathway accounted for a higher proportion in biological processes based on the Kyoto Encyclopedia of Genes and Genomes (KEGG) pathway enrichment analysis (Figure 4a, Appendix A). Furthermore, Gene Ontology (GO) enrichment analysis also found that these co-expressed genes participated in the ribosome pathway (Figure 4b, Appendix A). Therefore, we speculated that these *Pe14-3-3*s may be involved in the ribosome pathway in *P*. *edulis*. Further correlation analysis showed six of sixteen *Pe14-3-3* genes had a high correlation with most *Pe_ribosome*s in different height shoots of *P*. *edulis* (Figure 4c). Similarly, six of sixteen *Pe14-3-3*s also showed a great correlation with most *Pe_ribosome*s (Figure 4d) in the root of *P*. *edulis*. Finally, there were four *Pe14-3-3*s that had a strong correlation with eight *Pe_ribosome*s, both in the shoots and roots of *P*. *edulis* (Pearson’s correlation coefficients (PCC) > 0.75, *p*-value < 0.05)) (Appendix A). Furthermore, the intra-group correlation analysis of these twelve genes (four *Pe14-3-3*s and eight *Pe_ribosome*s) also showed a strong correlation (Appendix A, Appendix A).

### 2.5. Co-expression Network Construction and Validation of the Relationship between Pe14-3-3b/d and Pe_ribosomes

Based on the above analysis, a co-expression network was constructed with four *Pe14-3-3*s and eight *Pe_ribosome*s with a high correlation (Figure 5a). To further confirm how Pe14-3-3s played a role in the ribosome pathway, we checked the relationship between Pe14-3-3b/d and Pe_ribosome-1/4/5/6 using a targeted Y2H assay. As shown in Figure 5b, positive controls Pe14-3-3b/d and Pe_ribosome-1/5/6 co-transformed into yeast colonies could grow on both synthetic defined (SD)/-Leu/-Trp plates and synthetic defined (SD)/-Ade/-His/-Leu/-Trp/X-α-Gal plates, while the negative controls and the yeast colonies with other co-transformed genes only grew on the synthetic defined (SD)/-Leu/-Trp plates and did not grow on the SD/-Ade/-His/-Leu/-Trp/X-α-Gal plates. These results indicated that Pe14-3-3b/d could interact with Pe_ribosome-1/5/6 instead of Pe_ribosome-4 in yeast. Moreover, the qPCR results showed that *Pe14-3-3b*/*d* and *Pe_ribosome-1*/*5*/*6* had similar up-regulated expression trends with the increase in bamboo shoot height (Figure 5c): this further supported the positive relationship between *Pe14-3-3b*/*d* with *Pe_ribosome-1*/*5*/*6*. Therefore, we speculated that Pe14-3-3b/d could interact with Pe_ribosome-1/5/6 and they could work together to participate in the ribosomal metabolic pathway and affect the growth and development of *P*. *edulis*.

### 2.6. Ectopic Expression of Pe14-3-3b in Arabidopsis

To further elucidate the possible biological roles of *Pe14-3-3*s, *Pe14-3-3b* was selected for the recombinant overexpression (OE) vector construction. *Pe14-3-3b*, driven by *35S*, was transformed into *Arabidopsis*, meditated by *Agrobacterium tumefaciens*. A total of four hygromycin-resistant T3 plants were generated, of which three homozygous lines (OE-1, OE-2, and OE-3) were selected for further investigation. Under normal conditions, the root length of all transgenic lines was significantly longer than that of the wild type (WT) (Figure 6a–b), especially that of OE-3, wherein the mean root length increased by 58.94% compared with the WT after germination for nine days. Meanwhile, the bolting time of OE-1/2/3 was prior to that of the WT (Figure 6c), and the stem diameter of OE-1/2/3 was significantly larger than that of the WT (Figure 6d). The microscopic observation showed that OE-1/2/3 plants had larger cortical cells than those of the WT (Figure 6e). Additionally, the size of rosette leaves was bigger than those of the WT (Appendix A). Furthermore, qPCR analysis showed that the expression levels of *Pe14-3-3b* were significantly higher in OE-1/2/3 than that in the WT, especially in OE-3, which was 36.51 times of that in the WT. Similarly, compared with the WT, the expression levels of *AtRPL30e*, *AtRPL4* and *AtRPLS4A*, which were homologous genes of *Pe_ribosome-1/5/6,* respectively, showed higher expression levels in OE-1/2/3 than those in the WT (Figure 6f). These results indicated that overexpressing *Pe14-3-3b* increased the expression of *Pe_ribosome*s, resulting in a promotion of the growth and development of transgenic *Arabidopsis*.

## 3. Discussion

The 14-3-3 protein family, as a class of important factors for plant growth and development, have been widely studied in many eukaryotes [29]. Although the 14-3-3 gene family have been identified in multiple species, such as *O*. *sativa*, barley (*Hordeum vulgare*), soybean (*Glycine max*) and *Arabidopsis* [6,30,31,32], the molecular characteristics and evolution of 14-3-3 proteins in bamboo have not yet been studied. Therefore, the present study was conducted to identify the 14-3-3 gene family in *O. latifolia*, *P*. *edulis* and *B*. *amplexicaulis*, which represented diploid, tetraploid and hexaploid bamboo species. Based on the comprehensive analysis of the 14-3-3 gene family in bamboo, the function of *Pe14-3-3b* was further validated.

### 3.1. Characteristics Diversity of 14-3-3 Genes in Bamboos

In the present study, 36 *Pe14-3-3*s were identified in *P*. *edulis*, which was more than that identified in *O. latifolia* (8) and *B*. *amplexicaulis* (14) (Appendix A). This might be due to the fact that the *P*. *edulis* genome had undergone gene duplication events [21]. The physical and chemical characteristics of 14-3-3s in different bamboo species indicated that they were diverse. For example, the amino acid composition of some members of *Pe14-3-3*s was acidic and some were alkaline, while the amino acid composition of both *O*. *latifolia* and *B*. *amplexicaulis* was acidic. Likewise, *Pe14-3-3*s had both hydrophilic and hydrophobic members, but the members of both *O. latifolia* and *B*. *amplexicaulis* were hydrophobic (Appendix A). The structural diversity of exons/introns was generally considered to provide further insights into structural, evolutional and functional relationships [33]. In *P*. *edulis*, more than 60% of the members contained three or four introns (Appendix A). Similar results were found in *O*. *latifolia* and *B*. *amplexicaulis* (Figures S2a and S3a).

In addition, the number of exons and introns in different branches were variable (Appendix A, S2a), suggesting that evolution may be driving this structural diversity. This was also observed in *O*. *sativa*, *Medicago truncatula* and *G*. *max* [32,34,35]. Additionally, the intron lengths and arrangements were different in *Pe14-3-3*s, and similar results were also found in *O. latifolia* and *B*. *amplexicaulis*, further suggesting that the exon–intron structure could reveal the evolutionary diversity of the 14-3-3 gene family [33,36]. Furthermore, the conserved motifs of 14-3-3s in *O. latifolia*, *P*. *edulis* and *B*. *amplexicaulis* were highly variable (Figures S1b, S2b and S3b). The diverse motifs in the 14-3-3 family were the key structures of 14-3-3 proteins that could bind to many ligands [37], which directly affects the interaction of 14-3-3 proteins with other proteins to play a variety of functions.

### 3.2. Phylogeny and Evolution of 14-3-3 Proteins

Previously, genome duplication studies of 141 sequenced plant genomes demonstrated that, compared with *A. thaliana*, *P*. *edulis* was easier to evolve with *O*. *sativa* on shorter branches [28,38], and the relationship between *P*. *edulis* and *O*. *sativa*, *O*. *latifolia* and *B*. *amplexicaulis* was closer [39]. In our study, 37 14-3-3s from *O. latifolia*, *P*. *edulis* and *B*. *amplexicaulis* were clustered into ε and non-ε groups by phylogenetic analysis (Figure 1), which was consistent with previous reports on *Arabidopsis*, wheat (*Triticum aestivum*) and *G*. *soja* [7,30,40]. Our study also suggested the same phenomenon that most members of Pe14-3-3s, along with 14-3-3s in *O*. *sativa*, *O*. *latifolia* and *B*. *amplexicaulis*, clustered in close branches, while the evolutionary distance between *P*. *edulis* and *A. thaliana* was relatively far. It was crucial for us to infer the phylogeny and evolutionary direction of the 14-3-3 gene family between *P*. *edulis* and other plants (*O*. *sativa*, *O*. *latifolia* and *B*. *amplexicaulis*).

During evolution, plants undergo gene duplication events, among which whole-genome duplication and tandem duplication often promote the expansion of gene families [41,42], resulting in the diversity of gene function [43]. A previous study showed that *P*. *edulis* had undergone at least one round of whole-genome duplication, followed by multiple-segment duplication [21], which was considered to be the cause of the expansion of 14-3-3 family members in *P*. *edulis* during evolution. The differences of 14-3-3 gene members among three bamboo species might be mainly caused by the occurrence of genome-replication events [44], including series replication, such as fragment replication [45]. In our study, gene duplication and syntenic analyses of *14-3-3*s supported this assumption as 15 segmental duplication pairs were found within *Pe14-3-3*s, and 21 pairs were found between *Pe14-3-3*s and *Os14-3-3*s (Figure 2a). Similarly, 27 and 40 pairs were found between *Pe14-3-3*s and *Ol14-3-3*s, and *Ba14-3-3*s, respectively (Figure 2c,d). In addition, the *Ka*/*Ks* ratio can be used to indicate the selecting direction of one gene and measure the historical choice of coding sequences [44,45,46,47]. In our study, the *Ka*/*Ks* ratios (Figure 2d, Appendix A) showed that *Pe14-3-3*s had undergone a purification selection after duplication with limited functional divergence [27], which provided a better insight into the evolution of the 14-3-3 gene family.

### 3.3. Multiple Functions of 14-3-3s in Plant Growth and Development

The 14-3-3s are of vital importance during plant growth and development, which mainly manifests in reproductive and vegetative growth [4]. The function in reproductive and vegetative growth and expression patterns of *14-3-3*s have been studied in many plant species [48,49,50,51,52], but the possible function of *Pe14-3-3*s remains unclear. Gene expression patterns are an important manifestation of gene function. The expression patterns of *Pe14-3-3*s in various organs, and at different plant growth stages in moso bamboo (Figure 3), suggested that they had different functions, which was consistent with previous studies [53,54]. In line with our results shown in Figure 6a,b, the overexpression of *Pe14-3-3b* in *Arabidopsis* showed a longer root length than WT plants. Usually, shoot and stem diameter correspond to the number of cortical cells, and the increased number of cortical cells could improve shoot and stem growth [55]. The *Pe14-3-3b* transgenic *Arabidopsis* plants showed a larger stem diameter than WT plants (Figure 6d), which was supported by the larger size, instead of the number of cortical cells in transgenic *Arabidopsis* plants. Therefore, we deduced that *Pe14-3-3b* may improve shoot growth by increasing the size of cortical cells. The 14-3-3 protein family could also contribute to the development of leaves [51]. Similar results were found in our study (Appendix A): an overexpression of *Pe14-3-3b* in *Arabidopsis* led to the formation of bigger rosette leaves than WT plants, suggesting that *Pe14-3-3b* could accelerate the development of rosette leaves.

Furthermore, 14-3-3s could combine with multiple proteins to perform their functions in plant growth and development. Proteomic analyses revealed that 14-3-3 proteins were ribosomal-related [17]. Our study confirmed that Pe14-3-3b/d could interact with Pe_ribosome-1/5/6, among which Pe_ribosome-1 was the homologue of 40S ribosomal protein (AT5G58420) with an important role in seed germination and seedling transition [56]. The qPCR results showed that, with the increase in shoot height, the expression levels of *Pe14-3-3b/d* and *Pe_ribosome-1*/*5*/*6* all showed an up-regulated trend, consistent with previous studies, wherein 14-3-3 proteins could up-regulate the ribosome biogenesis [57]. Meanwhile, the expression level of *Pe_ribosome-4* showed a trend of increasing first, and then decreasing with the increase in shoot height, which may be a cause of its inability to interact with 14-3-3 proteins. Moreover, higher expression levels of *Pe14-3-3b* and *Pe_ribosome*s homologue genes were found in the transgenic *Arabidopsis* than those in WT plants (Figure 6f), which further suggested that these *Pe_ribosome*s work together with *Pe14-3-3b* to play an important role in the growth and development of moso bamboo. However, how the *Pe14-3-3b* is involved in ribosome pathways to regulate the development of moso bamboo shoots needs to be further studied.

## 4. Materials and Methods

### 4.1. Plant Materials

The moso bamboo (*Phyllostachys edulis*) samples were taken from the Jiangxi Academy of Forestry; the 15th internode was collected from representative shoots with heights of 0.5 m, 1.0 m, 4.0 m and 6.0 m. After treating with liquid nitrogen, the samples were stored in the refrigerator at −80 °C for further processing.

### 4.2. Identification, Characteristics, and Phylogenetic Analysis of 14-3-3 Family Members in Bamboos

The 14-3-3 protein sequences of *A*. *thaliana* and *O*. *sativa* (Appendix A) were used as queries against the genomes of *O*. *latifolia*, *P*. *edulis* and *B*. *amplexicaulis* in the database (http://bamboo. bamboogdb.org/, http://www.genobank.org/bamboo (accessed on 14 August 2021)) [58] with an *E*-value cut-off < 10^−10^. The HMM profile of the conserved domain of 14-3-3 (PF00244) was downloaded from the PFAM 32.0 database (https://pfam.xfam.org/ (accessed on 14 August 2021)) [59] and used to make sure that all protein sequences had 14-3-3 domain. The integrity of 14-3-3 conserved domain was determined using the online program SMART (http://smart.embl-heidelberg.de/ (accessed on 14 August 2021)) with an E-value < 0.1 [60]. The online ExPasy program (http://www.expasy.org/tools/ (accessed on 15 August 2021)) [61] was used to calculate the molecular characteristics of each 14-3-3 protein. The sequences of 14-3-3 proteins from *A. thaliana*, *B*. *amplexicaulis*, *O*. *latifolia*, *O*. *sativa and P. edulis* (Appendix A) were used to construct an unrooted phylogenetic tree according to the maximum-likelihood (ML) method (1000 bootstrap alignment replicates) by running MEGA 7.0 [62] with ClustalW alignment. Proteins were classified according to the distance of homology with 14-3-3 proteins in *A*. *thaliana* [2]. Previously known 14-3-3 protein sequences from *A*. *thaliana* (13) and *O*. *sativa* (8) used in this study were retrieved from the Phytozome v.12 database (http://phytozome.jgi.doe.gov (accessed on 14 August 2021)) [63]. Gene structure and motif analyses were performed using GSD 2.0 (http://gsds.cbi.pku.edu.cn/ (accessed on 20 August 2021)) [64] and the Multiple EM for Motif Elicitation program v5.4.1 (https://meme-suite.org/meme/ tools/meme/ (accessed on 20 August 2021)) [65], respectively, which were visualized using TBtools [66].

### 4.3. Scaffold Localization and Collinearity Analysis of 14-3-3 Genes

The scaffold distribution and syntenic relationships of the 14-3-3 genes of *P*. *edulis*, *O*. *sativa*, *O*. *latifolia* and *B*. *amplexicaulis* were obtained by MCScanX and visualized using TBtools [66]. The TBtools software was also used to calculate the *Ka* (non-synonymous substitution)/*Ks* (synonymous substitution) ratios of paralogous pairs to deduce selective pressure.

### 4.4. Expression Pattern Analysis of 14-3-3 Genes in Moso Bamboo

The expressions of 14-3-3 genes were analyzed by using the available transcriptome data generated from different tissues of moso bamboo, including rhizomes, rhizome buds, rhizome roots, roots at different growth stages (0.1 cm, 0.5 cm, 2.0 cm, 10 cm), shoots with different heights (0.2 m, 1.5 m, 3.0 m, 6.7 m), leaves and shoot buds (Sequence Read Archive accession number: SRS1847048-SRS1847073) [28]. We retrieved the fragments per kilobase per million (FPKM) values representing the expression levels of *Pe14-3-3s*, then a heatmap was drawn using the TBtools software [67] to display the expression profiles of *Pe14-3-3*s.

### 4.5. Co-expression, KEGG, GO and Correlation Analysis of Pe14-3-3s

The BambooNET (http://bioinformatics.cau.edu.cn/bamboo/index.html/ (accessed on 23 August 2021)) [67] was used to carry out a co-expression prediction analysis of *Pe14-3-3*s. Kyoto Encyclopedia of Genes and Genomes (KEGG) and Gene Ontology (GO) analyses were performed on an online website (https://www.omicshare.com/tools/ (accessed on 24 August 2021)) for genes co-expressed with *Pe14-3-3*s. Based on the RNA-seq data of moso bamboo in different tissues (roots with different lengths and shoots with different heights), the correlation analysis was conducted by calculating pairwise Pearson’s correlation coefficients (PCC) and *p*-values on an online website (https://www.omicshare.com/tools/ (accessed on 24 August 2021)) [68] between *Pe14-3-3*s and *Pe_ribosome*s: the results were visualized with TBtools [66]. In addition, a co-expression network was constructed using Cytoscape 3.7.1 [69] based on the results of the intra-group correlation analysis of *Pe14-3-3*s and *Pe_ribosome*s. PCC > 0.75 and *p*-values < 0.05 were set in the co-expression relationship analysis.

### 4.6. RNA Extraction and qPCR Analysis

The total RNA of the samples from moso bamboo was extracted using RNA Plus (Jianshi, Shanghai, China) following the instructions, and was reverse transcribed into cDNA with a PrimeScript™ RT Reagent Kit (TaKaRa, Kyoto, Japan) according to the manufacturer’s instructions. The qPCR experiments were performed using SYBR Green chemistry (Roche, Mannheim, Germany) on a qTOWER 2.2 system (Analytik, Jena, Germany) according to the manufacturer’s directions. *PeTIP41* was used as a reference gene for moso bamboo [70] to calculate the relative expression of the selected genes using the 2^−∆∆CT^ method [71]. The specific primers were designed by the Primer 5.0 software (Jin Wang, Soochow, China) (Appendix A).

### 4.7. Y2H Assay

Firstly, the full-length coding sequences of *Pe14-3-3b*/*d* and *Pe_ribosome-1*/*4*/*5*/*6* were obtained using specific primers (Appendix A) and cloned into pGBKT7 and pGADT7 vectors to form recombinant plasmids pGBKT7-Pe14-3-3s and pGADT7-Pe_ribosomes, respectively. According to the Yeast Protocols Handbook (Clontech, Mountain View, CA, USA), these combination constructs, including the positive controls of pGBKT7-53 and pGADT7-T, negative controls of pGBKT7-lam and pGADT7-T7 and the experimental groups of pGBKT7-Pe14-3-3s and pGADT7-Pe_ribosomes, were co-transformed into the yeast strain Y2HGold (*Saccharomyces cerevisiae*) and then plated on an SD solid-selection medium, including an SD/-Leu/-Trp medium and an SD/-Leu/-Trp/-His/-Ade/X-α-Gal medium, which were incubated at 30 °C until the appearance of colonies. Photographs were taken to record the growth of yeast colonies.

### 4.8. Transformation and Validation of Transgenic Arabidopsis Plants

The obtained CDS sequence of *Pe14-3-3b* was cloned into the overexpression (OE) vector Super1300 to form recombinant expression vector Super1300-*Pe14-3-3b*. The Super1300-*Pe14-3-3b* vector was introduced into the *Agrobacterium tumefaciens* strain GV3101 for *Arabidopsis* transformation using the floral dipping method [72]. Positive T1 transgenic plants were identified by PCR analysis and selected on a 1/2 MS solid medium with hygromycin (50 mg/mL); homozygous T3 seeds were used for subsequent experiments.

The seeds of wild-type (WT) *Arabidopsis* and transgenic *Arabidopsis* were surface-sterilized and seeded on a 1/2 MS solid medium. After vernalization, kept at 4 °C for two days, the seeds were transferred to an artificial climate chamber (temperature 23 °C, humidity 60–80%, light duration 16 h). About seven days later, the seedlings were transplanted into the nutrient soil (humus: vermiculite: 7:3) for further cultivation and subsequent experiment. For the measurement of root length, the seeds of transgenic lines and the wild type were transformed into vertical plates after germination for three days. The leaf phenotype was observed before bolting (around three weeks after germination). The stem diameter of all transgenic lines and WT plants were determined after germination for 30 days. Three technical and biological replicates were performed for all experimental data. The cDNA of transgenic lines and WT plants were used as templates for *Pe14-3-3b* expression analysis using *AtActin**2* as an internal reference [73]. The primer sequences are listed in Appendix A.

### 4.9. Statistical Analysis 

The statistical analyses were performed using IBM SPSS Statistics 22.0 (Armonk, NY, USA), and the mean and standard deviation of three biological replicates were presented. Significant differences were indicated at * *p* < 0.05, ** *p* < 0.01. For the measurement of the root length, 40 seedlings were statistically analyzed. For the measurement of the stem diameter, 30 seedlings were statistically analyzed. 

## 5. Conclusions 

In this study, we identified 8, 36 and 14 genes encoding 14-3-3 proteins from *O*. *latifolia*, *P*. *edulis* and *B*. *amplexicaulis*, respectively. The analyses of gene structures and conserved motifs indicated that *14-3-3*s were diverse. Differentially expressed patterns of *Pe14-3-3*s in different tissues of *P*. *edulis* supported their diversity. A co-expression network of *Pe14-3-3*s revealed that most *Pe14-3-3*s were ribosomal-related. Pe14-3-3b/d could interact with Pe_ribosome-1/5/6 in yeast two-hybrid assays, and they had similar expression patterns in shoots with different heights, which suggested that Pe14-3-3s worked together with Pe_ribosomes in the growth of *P*. *edulis*. Moreover, the overexpression of *Pe14-3-3b* resulted in the promotion of *Arabidopsis* growth, which further supported that *Pe14-3-3b* might play an important role in the rapid growth of *P*. *edulis*.

## Figures and Tables

**Figure 1 ijms-23-11221-f001:**
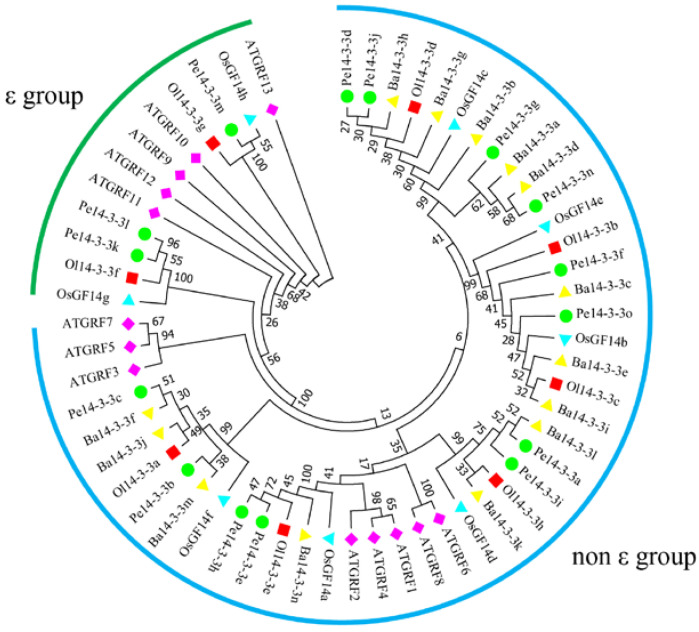
Phylogenetic tree of 14-3-3s in five plant species (At: *A*. *thaliana*; Ba: *B*. *amplexicaulis*; Ol: *O*. *latifolia*; Os: *O*. *sativa*; Pe: *P*. *edulis*). The phylogenetic tree was constructed via the maximum-likelihood (ML) method (1000 bootstrap replicates) using MEGA 7.0 software. The different numbers at nodes represent the bootstrap values for homologous genes. Purple prismatic: *A*. *thaliana*; yellow triangle: *B*. *amplexicaulis*; red square: *O*. *latifolia*; blue triangle: *O*. *sativa*; green circle: *P*. *edulis*; green and blue arcs: ε and non-ε groups.

**Figure 2 ijms-23-11221-f002:**
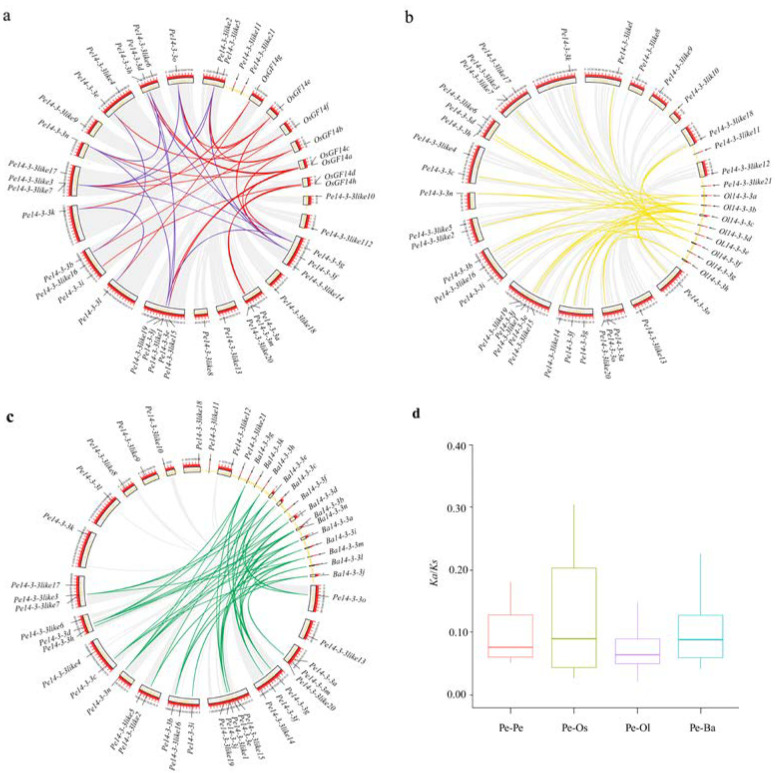
Collinearity of 14-3-3 genes in *P. edulis* and four other plants (Ba: *B. amplexicaulis*; Ol: *O*. *latifolia*; Os: *O*. *sativa*; Pe. *P*. *edulis*). (**a**) Collinearity among *Pe14-3-3*s (blue lines) and that between *Pe14-3-3*s and *Os14-3-3*s (red lines). (**b**) Collinearity of *Pe14-3-3*s and *Ol14-3-3*s (yellow lines). (**c**) Collinearity of *Pe14-3-3*s and *Ba14-3-3*s (green lines). (**d**) Boxplot statistics of *Ka*/*Ks* values of the 14-3-3 orthologous gene pairs within *P. edulis* and those between *P. edulis* and other three species.

**Figure 3 ijms-23-11221-f003:**
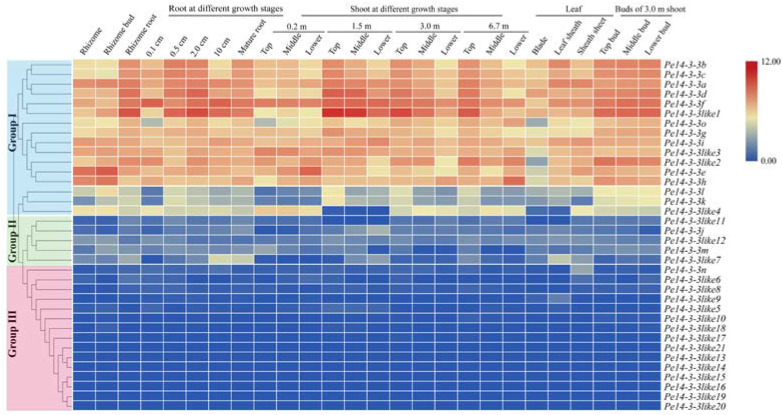
Expression analysis of *Pe14-3-3*s in different tissues of moso bamboo. Different color in map represents log_2_-based fragments per kilobase per million (FPKM) values as shown in bar at right of figure. *Pe14-3-3*s, divided into three groups on the basis of their expression levels in different tissues, are presented at the left and the names of different tissues are shown on the top.

**Figure 4 ijms-23-11221-f004:**
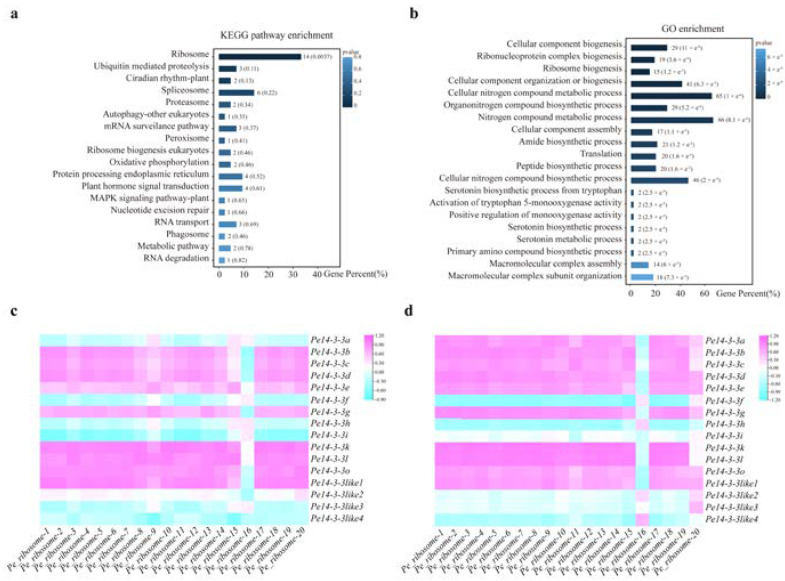
Function and correlation analysis of the genes co-expressed with *Pe14-3-3*s involved in the ribosome pathway. (**a**) KEGG pathway enrichment analysis of the genes co-expressed with 16 *Pe14-3-3*s. (**b**) GO enrichment analysis of the genes co-expressed with 16 *Pe14-3-3*s. Low *p*-values are shown in dark blue and high *p*-values are depicted in light blue. (**c**) The correlation analysis between 16 *Pe14-3-3*s with genes that participate in the ribosome pathway in the different height shoots of *P*. *edulis*. (**d**) The correlation analysis between 16 *Pe14-3-3*s with genes participated in the ribosome pathway in the roots of *P*. *edulis* with different lengths. The gradual change color from blue to purple indicates negative to positive correlation.

**Figure 5 ijms-23-11221-f005:**
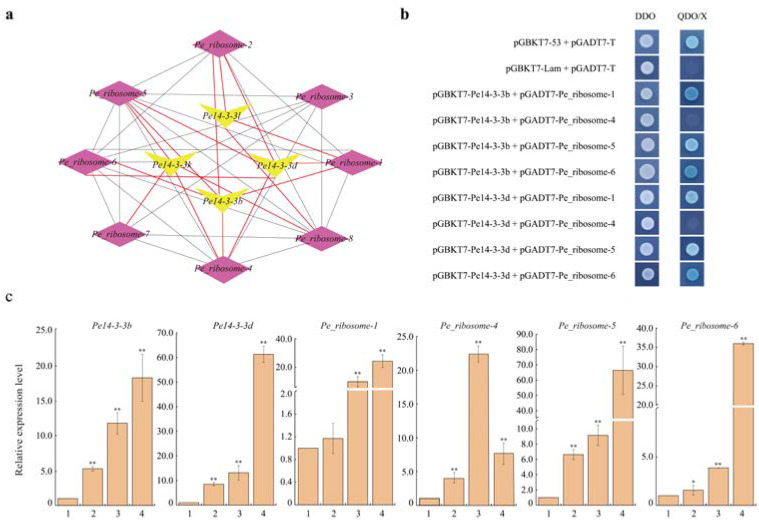
Relationship validation of Pe14-3-3s and Pe_ribosomes. (**a**) The co-expression network of four *Pe14-3-3*s with eight *Pe_ribosome*s, which had a strong correlation (PCC > 0.75, *p*-value < 0.05). The black and red lines indicate the co-expression relationship within *Pe_ribosome*s and those between *Pe_ribosome*s and *Pe14-3-3*s, respectively. (**b**) Y2H assay between Pe14-3-3b/d and Pe_ribosomes. Positive control, co-transformation with pGBKT7-53 and pGADT7-T; negative control, co-transformation with pGBKT7-Lam and pGADT7-T; DDO: SD/-Leu/-Trp, QDO/X: SD/-Ade/-His/-Leu/-Trp supplemented with X-α-Gal. (**c**) Expression analysis of *Pe14-3-3b*/*d* and four *Pe_ribosome*s in shoots with different heights (1: 0.5 m, 2: 1.0 m, 3: 4.0 m, 4: 6.0 m; the asterisks indicate the significant difference between the relative expression levels in different height shoots and that in 0.5 m shoots, * *p* < 0.05 and ** *p* < 0.01).

**Figure 6 ijms-23-11221-f006:**
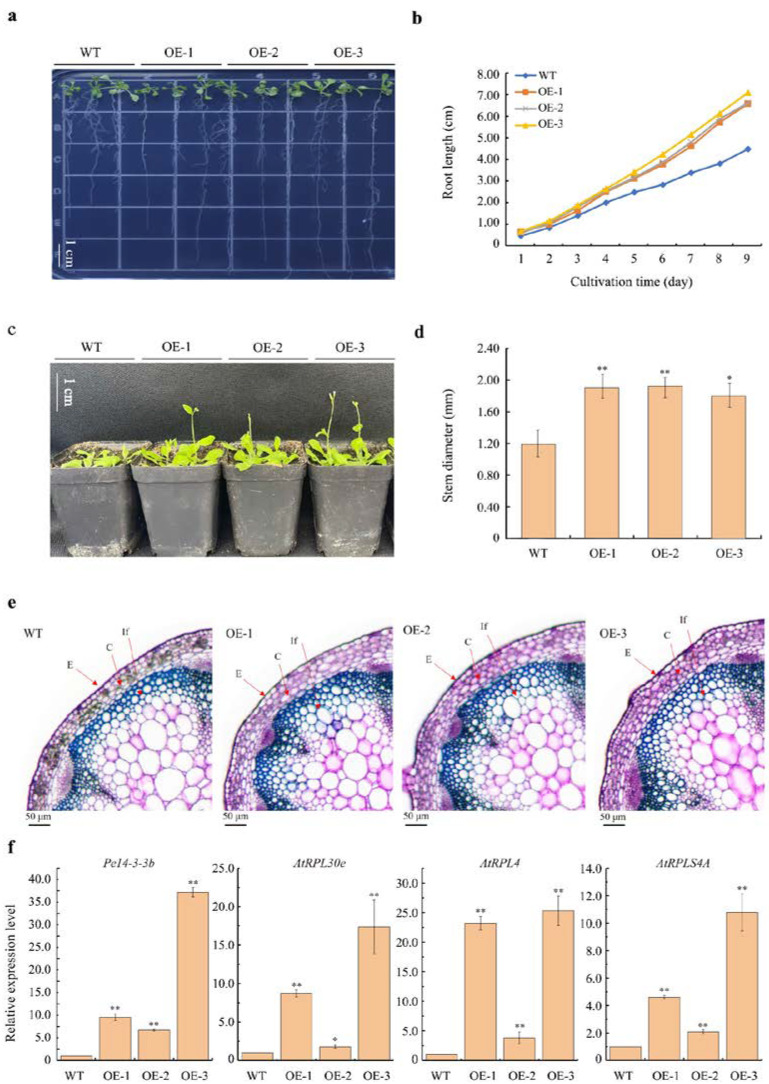
Ectopic expression of *Pe14-3-3b* in *Arabidopsis*. (**a**) Phenotype of seedlings on vertical plate after germination for 9 days of transgenic lines and WT plants. Bars = 1 cm. (**b**) Root length changing after germination on 1/2 Murashige and Skoog (MS) medium of transgenic lines and WT plants. The mean length was calculated from 40 individuals. (**c**) The bolting time was earlier in transgenic lines than that of WT plants (the pictures were taken around 3 weeks after germination, Bars = 1 cm). (**d**) Stem diameter analysis after germination for 30 days of transgenic lines and WT plants. (**e**) The microscope observation of apical stem of transgenic lines and WT plants. Bars = 50 μm; E: epidemics; C: cortex; If: interfascicular fibers. (**f**) Relative expression level of *Pe14-3-3b* and three ribosomal genes (*AtRPL30e*, *AtRPL4*, and *AtRPLS4A*) in transgenic lines and WT plants (asterisks indicate the significant difference levels (* *p* < 0.05, ** *p* < 0.01).

## Data Availability

All data are contained within the article and Appendix A.

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
