# Peer review of "Identification of the 14-3-3 Gene Family in Bamboo and Characterization of Pe14-3-3b Reveals Its Potential Role in Promoting Growth"

_ijms, 2022, doi:10.3390/ijms231911221_

Round 1
Reviewer 1 Report
The authors identify the 14-3-3 gene family in bamboo and characterize its functions in promoting growth. The authors investigate the molecular characteristics and evolutionary relationship of 14-3-3 isoforms. They deploy the co-expression networks and further apply yeast two-hybrid and ectopic expression to find that Pe14-3-3s worked together with Pe_ribosomes in the growth of P. edulis. Overall, the research is well planned and designed, and the diversity of techniques employed is sound. And the manuscript is well written and clear. However, I have the following comments on this work:
Line 49, it is well studied that 14-3-3 proteins play an important role in plant immunity. It might be better to add related information and citations to demonstrate the “diverse” function of 14-3-3 genes.
In figure 1, the figure legend is not clear. Could you provide detailed information on different colors and numbers?
Line 134, add a reference here.
Line 148-150, The statement is not clear. Do you mean roots/shoots at different growth stages, as you stated in the figure legend? If so, please be consistent throughout the manuscript.
Line 150, Based on the heatmap, in group I, Pel4-4-3b/c/a/d/f/like1 showed a higher expression level in most tissues than other isoforms. Moreover, they are highly expressed on the top of the roots, shoots, and buds. Although Pe14-3-3k and Pe14-3-3l showed similar patterns, the expression level is much lower than other isoforms in group I. In my opinion, Pel4-4-3b/c/a/d/f/like1 might be more involved in actively differentiated tissues.
Figure 3, the figure legend should be more elucidated by explaining the three groups and different plant tissues on the top.
For section 2.3, have you tried to confirm several gene expression levels by qPCR in addition to the prediction of published transcriptome data? And where did you get the published transcriptome data? Please also add the reference.
Line 166, Could you provide the reference or method to clarify how you found the 271 co-expression genes of the16 Pe14-3-3s (Group I family members)?
Figure 5b, negative controls between Pe14-3-3b/d and pGADT7-T might be more appropriate.
Discussion part: the discussion part is too long, with several repeatable sentences in the results section. I would suggest shortening the discussion part. For example, lines 132-134 and 310-312 are the same. Line 317-325, the function of 14-3-3s in plant growth and development has been stated in the introduction section. I suggest removing this paragraph or integrating it into the introduction section.

Reviewer 2 Report
Minor corrections are highlighted in the annotated PDF file of the manuscript. Sentences underlined in green may be rephrased for better clarity to the reader.
